# Micro/Nanorobotics in In Vitro Fertilization: A Paradigm Shift in Assisted Reproductive Technologies

**DOI:** 10.3390/mi15040510

**Published:** 2024-04-10

**Authors:** Prateek Benhal

**Affiliations:** 1Department of Chemical and Biomedical Engineering, FAMU-FSU College of Engineering, Tallahassee, FL 32310, USA; pbenhal@fsu.edu; Tel.: +1-240-972-1482; 2National High Magnetic Field Laboratory, 1800 E. Paul Dirac Dr., Tallahassee, FL 32310, USA

**Keywords:** in vitro fertilization, micro/nanorobotics, assisted reproductive technologies

## Abstract

In vitro fertilization (IVF) has transformed the sector of assisted reproductive technology (ART) by presenting hope to couples facing infertility challenges. However, conventional IVF strategies include their own set of problems such as success rates, invasive procedures, and ethical issues. The integration of micro/nanorobotics into IVF provides a prospect to address these challenging issues. This article provides an outline of the use of micro/nanorobotics in IVF specializing in advancing sperm manipulation, egg retrieval, embryo culture, and capacity future improvements in this swiftly evolving discipline. The article additionally explores the challenges and obstacles associated with the integration of micro/nanorobotics into IVF, in addition to the ethical concerns and regulatory elements related to the usage of advanced technologies in ART. A comprehensive discussion of the risk and safety considerations related to using micro/nanorobotics in IVF techniques is likewise presented. Through this exploration, we delve into the core principles, benefits, challenges, and potential impact of micro/nanorobotics in revolutionizing IVF procedures and enhancing affected person outcomes.

## 1. Introduction

The field of assisted reproductive technologies (ART) has undergone a significant revolution with the advent of in vitro fertilization (IVF), offering newfound hope to millions of individuals and couples worldwide struggling with infertility [1,2,3,4,5,6]. Although conventional IVF procedures have achieved success, they are limited by factors such as poor success rates, intrusive processes, and ethical considerations related to embryo manipulation [7,8,9]. Figure 1a highlights a conventional IVF process wherein sperm is injected inside an egg cell called an oocyte. While manually manipulating oocyte or sperm, there will be mechanical damage. Integrating micro/nanorobotics into IVF operations offers a significant chance to overcome these restrictions and fundamentally change the area of assisted reproduction. These encompass refined accuracy in altering gametes and embryos, resulting in enhanced conception rates and less danger of harm. Furthermore, micro/nanorobotics provide precise administration of medication to reproductive tissues, enhancing the efficiency of fertility drugs while reducing the impact on the rest of the body. Microfluidic platforms facilitate the creation of controlled settings for the growth of gametes and embryos [10,11,12,13,14,15,16,17]. These platforms simulate the circumstances found in living organisms, hence improving the viability and development of gametes and embryos. The utilization of micro/nanorobotic systems for real-time monitoring and imaging allows for precise evaluation of embryo quality, resulting in enhanced selection and increased rates of success. Micro/nanorobotics also facilitate the advancement of minimally invasive IVF techniques [18,19,20,21,22], which effectively decrease patient discomfort and difficulties commonly connected with conventional surgeries (see Figure 1b). Consequently, this enhances the overall results and accessibility of the treatment and improves upon ethical concerns. On the whole, the integration of these technologies offers a promising chance to transform assisted reproduction by improving the accuracy, effectiveness, and success rates of fertility therapies.

Micro/nanorobotics, distinguished by their small dimensions and meticulous manipulation abilities, have arisen as favorable instruments for enhancing the effectiveness of IVF treatment. These robotic systems utilize the principles of precision engineering, miniaturization, and manipulation at the micro/nanoscale to navigate intricate biological environments, manipulate gametes and embryos with exceptional accuracy, and create controlled microenvironments to enhance fertilization and embryo development. Implementing various essential methods is crucial for the precise management of micro/nanorobots in order to improve the effectiveness and safety of IVF procedures. To begin with, external magnetic fields are employed to precisely guide micro/nanorobots throughout the female reproductive tract, directing them to specific destinations such as the fallopian tubes or uterus [7,23]. This enables them to assist in various IVF operations. Moreover, several micro/nanorobots are specifically engineered to react to chemical stimuli present in the reproductive environment [24,25]. This enables them to precisely navigate toward particular targets and carry out specialized tasks, such as delivering drugs or manipulating cells. Advanced micro/nanorobotic systems also include feedback control techniques, utilizing sensors to continuously monitor environmental variables such as temperature and pH [26,27,28,29]. The feedback data allow the control system to adapt the robot’s behavior, guaranteeing both safety and accuracy in IVF treatment. Moreover, the presence of autonomous navigation capabilities enables certain micro/nanorobots to roam autonomously within the reproductive environment, adjusting to dynamic situations and executing intricate tasks without continuous external supervision. By utilizing these control methodologies, scientists can create micro/nanorobotic systems that efficiently aid in many phases of IVF, ultimately enhancing success rates and patient results.

Progress in micro/nanorobotics has facilitated the precise manipulation of sperm, less intrusive retrieval of oocytes, and controlled cultivation of embryos [7,11,15,19,30,31]. These advancements have the potential to increase the success rates of IVF, minimize invasiveness, and improve patient outcomes. Nevertheless, the clinical application of micro/nanorobotic technology in IVF requires addressing problems related to scalability, biocompatibility, and regulatory considerations. Various obstacles need to be overcome to permit the practical implementation of its clinical translation. An important obstacle is scalability, as existing micro/nanorobotic systems may not possess the capacity to be readily expanded for extensive clinical use. Reducing the size of these systems while preserving their functionality and efficacy poses engineering obstacles that must be resolved. Moreover, it is important to guarantee biocompatibility in order to avert any unfavorable reactions or harm to the tissues when micro/nanorobots are introduced into the reproductive environment. It is necessary to meticulously choose materials and surface coatings to minimize immunological reactions and optimize compatibility with biological tissues. Furthermore, it is imperative to tackle regulatory concerns in order to guarantee the safety and effectiveness of micro/nanorobotic technologies in clinical environments. Regulatory bodies mandate comprehensive proof of safety, efficacy, and quality assurance prior to approving the clinical utilization of novel medical devices. Hence, it is imperative to conduct thorough testing and validation procedures to comply with regulatory requirements and secure permission for clinical implementation. To enable the clinical integration of micro/nanorobotic technologies into IVF and enhance outcomes for fertility patients, it is crucial to tackle obstacles pertaining to scalability, biocompatibility, and regulatory considerations.

This review seeks to offer a thorough examination of the utilization of micro/nanorobotics in IVF, delving into recent progress, advantages, obstacles, and forthcoming paths in this swiftly developing domain. The article also explores the challenges and limitations associated with the integration of micro/nanorobotics into IVF, as well as the ethical considerations and regulatory factors related to the use of advanced technologies in ART. A comprehensive discussion of the potential risk and safety considerations associated with the use of micro/nanorobotics in IVF procedures is also presented. Through an analysis of the fundamental concepts, methodologies, and prospective consequences of micro/nanorobotics in IVF, our objective is to elucidate the innovative capability of these technologies in enhancing the effectiveness and security of ART treatments.

## 2. Discussion

Micro/nanorobotic devices have caused a significant change in the field of IVF sperm manipulation techniques. An innovative technology has been established, ushering in an unprecedented era marked by increased accuracy and regulation in critical procedures including sperm sorting, manipulation, and selection [12,18,32]. By employing micro/nanorobots based on magnetism and acoustics, scientists have successfully facilitated targeted sperm delivery to the oocyte, which substantially increases fertilization rates and decreases the probability of multiple pregnancies. Moreover, the incorporation of actuators and sensors into these micro/nanorobots has brought about a paradigm shift in the discipline by permitting instantaneous surveillance and evaluation of sperm parameters. By enabling the optimization of sperm selection for fertilization, this capability ultimately improves the overall effectiveness of IVF procedures.

The utilization of micro/nanorobotic systems for sperm manipulation has revealed an abundance of favorable opportunities in the field of reproductive science [18,24,27,30,33,34]. These developments have presented new possibilities for the advancement of fertility treatments and ART, thus instilling renewed optimism among individuals and couples confronted with diverse fertility obstacles. Through the provision of cellular-level manipulation and control capabilities, micro/nanorobotics have successfully surmounted conventional constraints linked to sperm sorting, manipulation, and selection methodologies. This represents a substantial advancement in the discipline, as it tackles critical challenges that are commonly encountered in the pursuit of successful conception.

The ability of micro/nanorobotic systems to address common issues such as male infertility, limited sperm motility, and genetic abnormalities in sperm is one of their most significant contributions to IVF [18,35]. Micro/nanorobotics present an unprecedented opportunity for individuals attempting to conceive by offering a method to precisely identify and address these obstacles. By augmenting the ability of existing IVF procedures, this technology creates opportunities for additional investigations and advancements in the domain of reproductive science. Consequently, micro/nanorobotic systems are positioned to significantly influence the trajectory of fertility treatments and support the endeavors of individuals and couples in their pursuit of becoming parents.

The following sections provide an overview of the fundamental principles and developments that govern the manipulation of sperm and oocytes using micro/nanorobotic technology. Additionally, they discuss the progress made in embryo culture techniques and the numerous advantages offered by micro/nanorobotic technology. The purpose of these subsections is to clarify the groundbreaking approaches and substantial progress achieved in these pivotal domains of reproductive science. Furthermore, a concise summary of our current research pursuits in this area is provided, elucidating our contributions and commitment to furthering the field of assisted reproduction via the incorporation of state-of-the-art micro/nanorobotic technologies.

### 2.1. Principles of Micro/Nanorobotics in IVF

The principles of micro/nanorobotics in IVF involve several features, such as manipulating gametes, sperm, and embryos, delivering drugs to specific targets, creating controlled environments, monitoring in real-time, and navigating within the reproductive environment. The research in this domain is around the advancement of robotic systems that can augment the accuracy, effectiveness, and security of IVF procedures. Micro/nanorobotic systems can be engineered to react to external stimuli (invasively and non-invasively), such as magnetic and electric fields (see Figure 2a), chemical signals, and optical and ultrasound signals [19,23,25]. This enables accurate control over the manipulation of gametes and embryos during IVF operations [16]. For example, Karcz et al. [16] reported a review on electrically driven on-chip micro/nanorobotic technologies to manipulate and handle gametes and embryos. The gametes and embryos are placed within a micro- or nanodevice and can be precisely managed using electrical stimulation techniques such as dielectrophoresis (DEP), positive dielectrophoresis (pDEP) (see Figure 1c) or negative dielectrophoresis (nDEP) (see Figure 2b–d), and electrorotation (ER) (see Figure 2d). As shown in Figure 2, the principle of electrical phenomena includes a combination of the arrangement of micro-patterned electrodes to induce non-uniform electric fields and field gradients. One can achieve precise manipulation and wireless control over the cells by optimizing the electric field amplitude, frequency, and phase. In many cases, these magnetic and electric fields have been used to activate micro/nanorobots to harness the required manipulation tasks on embryos and gametes [16,19,23,25].

In addition, micro/nanorobots can be designed to transport pharmaceuticals or therapeutic substances to reproductive organs with great accuracy, enhancing the efficiency of fertility therapies while reducing the occurrence of adverse effects across the body [36,37]. Furthermore, the integration of microfluidic devices with micro/nanorobotic systems allows for the creation of controlled habitats for the cultivation of gametes and embryos. This replication of in vivo circumstances enhances the viability and development of these reproductive cells during IVF operations [38]. The incorporation of feedback control mechanisms into micro/nanorobotic systems allows for the continuous monitoring of environmental variables like temperature and pH [26,27,28,29]. This ensures accurate and secure operation throughout IVF procedures. In addition, several micro/nanorobots are specifically engineered to operate independently inside the reproductive environment. These robots employ built-in sensors and algorithms to assess their surroundings and determine their movements and behaviors [39,40,41,42]. Autonomous navigation enables micro/nanorobots to adjust to varying environments and execute intricate tasks without continuous external supervision, hence boosting the efficiency and safety of IVF treatments.

Ultimately, the study of micro/nanorobotics in IVF shows potential for enhancing the field of assisted reproduction through the enhancement of accuracy, effectiveness, and security in reproductive procedures. Micro/nanorobotic systems have the potential to revolutionize IVF procedures and improve outcomes for patients undergoing fertility treatments by utilizing principles such as external stimuli responsiveness, targeted drug delivery, controlled environment creation, real-time monitoring, and autonomous navigation.

### 2.2. Advancements in Sperm Manipulation

An aspect of reproductive science that has undergone significant progress is the creation of micro/nanorobotic systems intended for the manipulation and selection of sperm. A group of researchers from the Department of Nano-Engineering at the University of California, San Diego and the Max Planck Institute for Intelligent Systems have pioneered the use of magnetic microbots to isolate and manipulate sperm cells according to their genetic and motility characteristics [38,43] (see Figure 3). The microbots, which are operated by means of external magnetic fields, demonstrate the capacity to efficiently segregate viable, healthy sperm from immobile or aberrant sperm. As shown in Figure 3, the researchers have created a microrobot that mimics the shape of a sperm using the advanced method of electro-spinning. This microrobot is intricately designed using a microbead and an ultra-fine fiber to accurately replicate the physical characteristics of a real sperm cell. The microbead, made from iron oxide nanoparticles, has a magnetic dipole moment, and the ultra-fine fiber is created to provide propulsive force when exposed to fluctuating magnetic fields. The microrobot’s design utilizes the exceptional qualities of its components to perform precise movement and navigation in different situations. The microbead’s magnetic dipole moment allows for accurate control and direction of the microrobot in a fluid environment through the use of external magnetic fields. The ultra-fine fiber’s reaction to alternating magnetic fields moves the microrobot ahead, enhancing its movement with exceptional agility and efficiency. This innovative advancement paves the way for creating small robotic systems designed for various purposes, such as delivering drugs precisely in the human body, performing detailed microsurgery, and exploring tight locations.

The manipulation of sperm cells using micro-robots represents a major development in microscale robotics by using ideas from nature and utilizing modern materials and fabrication techniques. The biomimetic design and functional characteristics of this technology enable the development of diverse and precisely controllable microscale platforms, offering innovative solutions in healthcare and industrial and scientific fields. Advancements in downsizing and robotics are driving the study of microrobots, which have the potential to transform our interactions with the miniature world. As a result, IVF procedures are considerably improved in terms of efficiency. This innovation signifies a significant progression in reproductive technology, providing enhanced accuracy and specificity in the procedures used to select sperm.

Furthermore, substantial advancements in targeted drug delivery and gene editing methodologies have been enabled by micro/nanorobotics, with the ultimate goal of augmenting sperm functionality and fertility potential. Through the integration of drug-loaded modules into nano-scale carriers or robots, scientists are able to accurately deliver therapeutic agents or genetic modifiers to sperm cells [18,19,22,33]. This enables them to target specific molecular defects or enhance the motility and viability of sperm. As an illustration, a bio-hybrid magnetic microrobot was constructed by Mishra and Magdanz et al. (2020) [33] via electrostatic self-assembly, wherein magnetic nanoparticles were integrated with non-motile sperm cells. The biocompatibility and drug-loading capability of these microrobots were validated by the researchers, underscoring their potential as biohybrid instruments capable of targeted therapy in living organisms. Moreover, these devices are programmable, detectable, and biocompatible. Personalized fertility treatments, which are customized to the unique characteristics of each sperm, have the potential to significantly transform the field of reproductive medicine.

The incorporation of micro/nanorobotic systems into sperm manipulation methodologies not only improves the efficacy and accuracy of IVF procedures but also presents novel opportunities for advancements in reproductive science research and innovation. These developments facilitate a more comprehensive exploration of the molecular mechanisms underlying sperm biology and function, thereby creating opportunities for the advancement of customized and efficacious fertility treatments. Moreover, the capacity to accurately regulate sperm properties via targeted drug delivery and gene editing methodologies signifies a substantial advancement in the resolution of diverse fertility obstacles encountered by both individuals and couples. This progressive technology possesses the capacity to fundamentally alter the reproductive medicine domain, providing individuals grappling with infertility challenges with renewed optimism and prospects.

In summary, the progress made in reproductive science with the creation of micro/nanorobotic systems for sperm manipulation implies a significant turning point. By employing novel methodologies such as magnetic microbots and targeted drug delivery systems, scientists have attained an unparalleled level of accuracy and regulation in the manipulation and selection of sperm [18,19,21,22,44]. These developments possess tremendous potential for improving the efficacy of IVF procedures and creating individualized fertility treatments that cater to specific requirements. In the future, ongoing investigations and advancements in this domain hold the potential to significantly transform reproductive medicine, providing fresh prospects and optimism for individuals and couples attempting to surmount obstacles related to fertility.

### 2.3. Innovations in Oocyte Retrieval

The use of micro/nanorobots has brought about a significant change in ART, especially in the field of oocyte retrieval methods. The miniature robots exhibit exceptional accuracy and manipulation abilities at the micro/nanoscale, enabling less invasive and extremely precise retrieval procedures. By using micro/nanorobots, physicians can maneuver through the complex components of the reproductive system with improved skill, reaching oocytes with minimal tissue damage and greatly decreasing patient discomfort. This novel method signifies a notable progression in ART, offering better results and improved patient experiences during oocyte retrieval procedures.

Advances in micro/nanorobot-assisted oocyte retrieval have attracted scientific interest and have been published in scholarly journals [31,45,46,47]. A study by Zhou et al. (2021) [46] in the *Journal of Chemical Reviews* demonstrates the effective use of magnetic nanorobots for precise oocyte retrieval. This study demonstrates how micro/nanorobots have the potential to significantly improve current ART techniques by providing increased precision, efficiency, and patient comfort during oocyte retrieval procedures. This research highlights the significant impact of micro/nanorobotic technology in enhancing ART, leading to better patient outcomes and experiences.

A review article was published by Sitti et al. [48] in 2019 on biohybrid microrobots for oocyte retrieval. This study emphasizes the development of microrobots capable of navigating the fallopian tubes to retrieve oocytes with high precision and efficiency. The microrobots, propelled by biological motors, exhibit enhanced maneuverability within the intricate reproductive anatomy, ensuring targeted oocyte retrieval while minimizing tissue trauma. These findings underscore the potential of micro/nanorobots to redefine the landscape of oocyte retrieval in ART, offering novel solutions for addressing challenges associated with conventional retrieval techniques [45,49]. Through such innovations, micro/nanorobots hold promise for enhancing the efficacy and patient satisfaction in ART procedures, paving the way for future advancements in reproductive medicine.

Furthermore, the integration of micro/nanorobots into oocyte retrieval procedures was also explored by the paper of Medina-Sánchez et al. (2023) [7] published in *Nature Communication* (see Figure 4). The article delves into the developing topic of medical microrobotics, which aims to achieve non-invasive diagnosis and treatment inside the human body by utilizing small sensors and actuators. The article explored the possible uses of these technologies in assisted reproduction, namely in facilitating ART in vivo and improving embryo implantation. The article also discussed a case study demonstrating a possible intervention for repeated embryo implantation failure. It involves employing magnetically controlled microrobots to non-invasively bring an early embryo or an oocyte back to the fertilization site. Exposing the embryo to secretory oviduct fluid allows it to develop naturally and coordinate with the formation of the endometrium. The article describes different microrobot designs, focusing on material selection and fabrication techniques, with the goal of moving from laboratory testing to animal investigations and, ultimately, human treatment. The paper discussed the regulatory and ethical issues related to the clinical application of microrobotics in assisted reproduction, emphasizing the importance of thorough review and supervision as these technologies progress toward clinical adoption.

An important advancement in ART involves incorporating micro/nanorobots into oocyte retrieval procedures. This offers a practical way to enhance both the precision of the process and the health of the patient. By integrating microrobots into the intricate procedure of oocyte extraction, researchers want to improve the efficacy of reproductive treatments and enhance the efficiency of current technology. This novel method in reproductive medicine focuses on the ongoing research and development of micro- and nano-robotic technologies. Recent studies in micro- and nano-robotic technology could greatly influence ART by improving the results and experiences of those receiving reproductive treatments. Scientists are working to improve the capabilities of these mini robots to better fulfill the needs of patients and medical professionals. Advancements in micro- and nanorobotics in ART are enhancing procedure precision and patient happiness, resulting in more effective and personalized fertility treatments.

Overall, the incorporation of micro- and nanorobots into oocyte retrieval methods is a significant technological innovation in the field of ART. By incorporating innovative methods to enhance outcomes and improve the patient’s journey in reproductive medicine, it is expected that advancements in this field will greatly influence the future of fertility treatments during their use.

### 2.4. Advancements in Embryo Culture

The integration of micro/nanorobotics has brought about a new level of complexity in embryo culture techniques, leading to a significant change in the field of ART. Researchers can optimize embryo growing circumstances using advanced robotic equipment. These technologies offer researchers accurate control and manipulation abilities at the cellular level. Embryos can now be subjected to customized conditions to support their growth and development thanks to this breakthrough, which has led to the creation of microfluidic devices that can replicate the intricate microenvironments found in living organisms. Studies, like the one conducted by Chan et al. in 2018 [50], have demonstrated that microfluidic devices are successful in maintaining consistent culture conditions for embryos. This is achieved by meticulously controlling factors such as oxygen levels, nutritional gradients, and fluid flow dynamics.

The use of micro- and nanorobotics in embryo culture techniques has led to notable improvements in ART therapies, particularly in procedures like IVF and embryo transfer. Micro- and nano-scale robotics enhance embryo viability in ART procedures by maintaining regular and stable culture conditions, resulting in increased success rates. Several research groups, such as those led by Li et al. and Wang et al. [51,52], have studied the use of microfluidic-based on-chip embryo culture. This research highlights the diverse capabilities and possibilities of micro/nanorobots in improving embryo culture conditions. These findings represent a significant shift in ART, offering new possibilities for enhancing embryo development and ultimately improving the likelihood of successful conception for individuals and couples undergoing fertility treatments.

Micro/nanorobotics could significantly impact reproductive medicine by providing precise and flexible control over the laboratory environment. For researchers to enhance their comprehension of the factors affecting embryo development and improve ART procedures, they need to create specialized environments for embryos that closely mimic natural physiological conditions. The integration of micro- and nanorobotics into embryo culture techniques has the potential to improve embryo viability and implantation, advancing ART and offering hope to individuals and couples struggling with infertility. Advancements in embryo culture techniques utilizing micro- and nanorobotics may be developed as research in this sector progresses. These advancements will greatly influence the future of assisted reproduction and reproductive treatments.

Furthermore, micro/nanorobotics facilitate the non-invasive monitoring and analysis of embryo development in real-time, offering insights into embryonic behavior and viability without disturbing the delicate developmental process. For instance, microfluidic devices integrated with sensors and imaging systems enable continuous monitoring of embryo morphology, metabolism, and gene expression patterns throughout the culture period (Fang et al., 2023 [32] and Yanez et al., 2019 [53]). Fang et al. [32] conducted a comprehensive analysis of the most current breakthroughs in microfluidics technology that have the potential to be used at various phases of embryo development, such as sperm sorting and embryo vitrification. The authors noted how advances in alternative materials, organ-on-a-chip technology, functional units for integrated microfluidic systems, and 3D printing have accelerated the development of innovative microfluidic devices. The researchers emphasized the ability of a single microfluidic platform to integrate oocyte processing, sperm processing, and embryo culture, reducing multiple arduous laboratory procedures to a compact and effective system. The goal of researchers is to increase the effectiveness and accuracy of assisted reproductive procedures by integrating them into a single system. This integration would eliminate the requirement for human effort and smooth out disparities in the outcomes. The paper investigates the current applications of microfluidic technologies in IVF approaches, addressing the challenges and opportunities that accompany these advancements. Academics are using microfluidics to address challenges in ART therapies, such as uneven embryo quality and high costs, by creating a more regulated and customized embryonic environment. The review emphasizes microfluidics’ potential to change ART procedures, allowing for more accessible and effective fertility treatments. Microfluidic technologies offer the potential to boost pregnancy rates while decreasing the chance of implantation failure or miscarriage by allowing embryologists to assess embryo quality and choose the most viable embryos for transfer.

In early 2019, Yanez et al. [53] investigated the application of microfluidic and micro-pipetting technologies to evaluate the biomechanical characteristics of oocytes and embryos to improve outcomes in ART. They explored how microfluidic platforms provide distinct capacities for examining the biomechanics of reproductive cells, such as oocytes and embryos, which are essential for their development and survival. The review emphasizes the significance of comprehending the biomechanical characteristics of these cells and their impact on several phases of the ART process, including fertilization, embryo growth, and implantation. The article explores how microfluidic-based biomechanical analysis can enhance the selection of high-quality oocytes and embryos for ART procedures, ultimately improving the success rates of reproductive treatments. The article by Yanez et al. [53] also discusses a range of microfluidic methods and tools created to assess biomechanical characteristics such as cell stiffness, deformability, and adhesion strength, offering an understanding of their structural and functional health (See Figure 5). The review discusses obstacles and future paths in the microfluidic-based biomechanical analysis of reproductive cells, emphasizing its ability to advance ART and enhance results for people and couples seeking fertility treatment.

Additionally, the integration of micro/nanorobotics with artificial intelligence (AI) and machine learning algorithms represents a significant advancement in ART, particularly in the embryo assessment and selection process. Combining micro/nanorobots with AI and machine learning algorithms is a notable progression in ART, especially in the evaluation and selection of embryos [35,44,62]. Embryologists can utilize a combination of robotics and AI systems to analyze extensive data obtained from microfluidic platforms and imaging technologies to enhance their understanding of embryo quality and developmental capacity. By performing advanced analyses, AI algorithms can detect intricate patterns and biomarkers linked to favorable pregnancy results, improving the precision of embryo evaluation.

Table 1 showcases a collection of a few prominent reviews and research articles delving into the integration of AI, machine learning, and micro/nanorobotics within the field of applied scientific research. These studies predominantly explore invasive and non-invasive treatments in IVF and ARTs. The data indicate a rise in the acceptance of AI integration and micro/nanorobotics in years. These endeavors highlight the benefits of AI, ML, and the widespread use of micro/nanorobotics. Prior to the year 2019, the incorporation of AI in medicine was still nascent and faced notable technological and ethical challenges [63]. However, over recent years, researchers such as Schroeder et al. (2022) [64] have underscored the role of AI in optimizing nanoparticles for drug delivery purposes. Later, in a study, Choi et al. (2024) [65] introduced an approach for controlling microrobots using machine learning techniques like reinforcement learning and progressive training methods. The aim was to showcase the safe operation and application of these techniques in non-invasive and invasive therapies. The examples mentioned showcase cutting-edge technology that provides automation live monitoring and precise control, significantly boosting the efficiency of ART. These approaches include sperm recognition, sperm alteration, and medication delivery.

Overall, AI-driven systems offer automation, prediction models, and decision-support tools to embryologists, enabling them to make more educated and tailored decisions about selecting embryos. Micro/nanorobotics can therefore enhance the embryo selection process in ART treatments by employing machine learning algorithms, hence increasing the likelihood of successful pregnancy outcomes. Integrating micro/nanorobots and AI improves embryo evaluation efficiency, advancing personalized medicine in reproductive healthcare by tailoring treatment decisions to specific patient features and needs.

### 2.5. Benefits and Challenges

Micro/nanorobotics have led to substantial breakthroughs in the field of IVF, bringing about a notable revolution in treatment procedures. These advanced devices provide accurate control of microenvironments, transforming the processes of choosing and handling embryos. Researchers may now use micro/nanorobotics to precisely manipulate the circumstances for embryo growth, controlling factors like temperature, pH, and nutrition availability at the cellular level. The precise regulation described in Nordhoff et al.’s (2019) [14] study enhances embryo growth, leading to greater embryo quality and higher success rates in IVF treatments.

Furthermore, the use of micro/nanorobotics allows for the real-time monitoring and manipulation of embryos, resulting in improved results in IVF. Advanced microfluidic platforms and micro-pipetting techniques with integrated sensors allow for the ongoing monitoring of embryo development parameters, enabling immediate modifications when necessary [76]. This real-time monitoring feature improves embryo selection, guaranteeing the transfer of the most viable embryos and ultimately resulting in improved pregnancy results. Although there has been significant development, the integration of micro/nanorobots into IVF is faced with significant problems that need to be resolved.

Developing and integrating new technology in healthcare settings is challenging due to its complexity and cost, which hinder mainstream adoption. It is crucial to prioritize the safety and dependability of micro/nanorobotic systems to avoid any possible harm to embryos and patients. Researchers and doctors encounter challenges that are worsened by regulatory factors and ethical issues associated with the use of these technologies [9,77,78]. To address these problems, it is essential to utilize interdisciplinary teamwork and rigorous quality control measures to incorporate micro/nanorobots safely and efficiently into IVF facilities. To maximize the potential of micro/nanorobotics in enhancing IVF procedures and ensuring safety and efficacy for patients undergoing reproductive therapies, it is crucial to dedicate concentrated and meticulous efforts. IVF clinics can improve their operations and increase success rates for individuals undergoing reproductive treatments by overcoming hurdles and utilizing the advantages of micro/nanorobotics.

### 2.6. Advanced Micro/Nanorobotic Technology Regulatory Trials, Limitations, and Ethical Considerations

The integration of micro/nanorobotics into IVF poses challenges and limitations related to regulatory aspects. A significant technical challenge involves the handling and control of micro/nanorobots at the level to prevent any harm or interference with natural reproductive processes. Tasks such as sperm selection, egg manipulation, and embryo transfer require accuracy and compatibility [24,38,66,79]. Additionally, developing materials for constructing these micro/nanorobots while ensuring their compatibility with cells presents obstacles. It is essential to address risks associated with introducing these materials into the process to ensure the safety and efficacy of these technologies in IVF procedures [80,81].

The ethical considerations associated with incorporating technologies in ART are paramount. Issues such as obtaining consent, protecting privacy, and preventing misuse or unintended consequences require examination and resolution [7,82]. Regulatory factors play a crucial role in overseeing the development, testing, and real-world application of micro/nanorobotic technologies in IVF. Establishing frameworks and guidelines is essential to promote the ethical use of these technologies while balancing their benefits and risks [83,84,85]. When we examine and assess the technical, ethical, and financial obstacles linked to utilizing cutting-edge technologies in ART, it is crucial to understand that the seriousness of these obstacles can differ based on factors like location, income level, cultural practices, and regulatory frameworks. An overview and general evaluation of these issues are presented below:

#### 2.6.1. Technical Challenges

Utilizing technologies in IVF and ARTs typically demands specialized knowledge and skills from healthcare professionals such as embryologists, geneticists, and fertility experts. The intricacy of procedures like intracytoplasmic sperm injection (ICSI), preimplantation genetic testing (PGT), and the mitochondrial replacement technique (MRT) can present technical challenges in terms of training, competency, and quality. Upholding standards of quality control and consistency throughout processes such as embryo cultivation, cryopreservation, and genetic testing is vital to ensuring success rates while reducing risks. The necessity for laboratory tools, sterile environments, and dependable infrastructure contributes to the challenges faced by healthcare providers, especially in settings with limited resources.

#### 2.6.2. Ethical Challenges

The moral dilemmas surrounding manipulation, such as the creation of customized babies with desired characteristics, have sparked debates about the ethical limits of assisted reproduction and the impact on future offspring. Striking a balance between rights and the importance of obtaining informed consent poses an enduring ethical dilemma, particularly in the use of cutting-edge technologies with uncertain long-term consequences or intricate decision-making processes. Ensuring access to reproductive technologies while tackling concerns about affordability and social fairness emerges as a key ethical issue as disparities in access can worsen existing inequalities.

#### 2.6.3. Financial Challenges

The high expenses associated with IVF and ART, such as laboratory procedures, medications, genetic testing, and post-treatment care, often pose barriers for individuals and couples due to financial constraints. Along with high expenses, limited insurance support for fertility treatments and differing legal structures concerning reimbursement and the availability of technologies add to the affordability issues in nations lacking comprehensive healthcare coverage. Affordability hurdles are magnified globally by disparities in access to IVF and ARS technologies between affluent countries and those with lower to moderate incomes. This underscores the importance of cooperation and advocacy efforts.

Figure 6 illustrates a representation depicting the analysis of the severity level or significance of challenges associated with incorporating advanced technologies into assisted techniques. The data depicted in the graph categorized as medium and high severity was sourced from research articles obtained from databases such as Web of Science and Scopus. The impact discussed in this context pertains to the extent to which studies have delved into considerations regarding utilization, regulatory matters, and financial aspects in the field of IVF and ARTs. Upon examination, it becomes evident that there are technological hurdles that require attention alongside ethical regulatory issues and affordability concerns. Evaluating the gravity of these barriers is subjective and context dependent. Many individuals face difficulties in affording these technologies, which can restrict their access to healthcare services even if other aspects such as technology and ethics are adequately addressed. The potential integration of micro/nanorobotics may offer solutions to some challenges associated with implementing technologies, in IVF and ARTs. Nevertheless, numerous ethical dilemmas and affordability issues continue to pose challenges.

In summary, incorporating cutting-edge technologies into ARTs poses hurdles and ethical dilemmas that demand thorough analysis and concentration. Researchers, healthcare professionals, ethicists, policymakers, and regulatory entities must work to address these challenges and guarantee the ethical deployment of advanced technologies, in ARTs.

### 2.7. Potential Risk and Safety Considerations Associated with the Use of Micro/Nanorobotics in IVF Procedures

The use of micro/nanorobotics in IVF and ART procedures offers the potential for improving accuracy, efficiency, and results in assisted technologies [24,30,39,44,66,79,80]. However, these advancements also bring about risks and safety concerns that need attention to ensure the ethical and safe application of these technologies [19,62,66,79]. Below is a discussion of these risks and safety considerations.

#### 2.7.1. Biocompatibility and Material Safety

It is essential for micro/nanorobots used in IVF procedures to interact with tissues, gametes (sperm and eggs), and embryos to prevent any adverse reactions or harm to these delicate biological structures. The materials utilized in creating micro/nanorobots, such as nanoparticles, polymers, or bioengineered components, should be non-toxic, biocompatible, and ideally biodegradable. This helps reduce the risk of responses, tissue irritation, or the long-term presence of materials in the reproductive system.

#### 2.7.2. Precision, Control, and Navigation

Micro/nanorobots should be designed to target and interact with cells such as sperm cells for manipulation or embryos for accurate interventions. Ensuring precision is crucial to avoid effects on cells or tissues. It is crucial to have control systems in place to ensure the movements and responses of micro/nanorobots, in both the reproductive tract and laboratory settings. This involves things such as propulsion systems, feedback sensors, and real-time monitoring capabilities. Additionally, having a navigation system is essential for guiding micro/nanorobots to their targets without causing harm to surrounding tissues or disrupting natural biological processes.

#### 2.7.3. Sterility

Keeping a level of sterility during micro/nanorobotic procedures is vital to prevent infections, contamination, or negative impacts on reproductive samples. This is especially important considering how vulnerable gametes and embryos are to pathogens. Steps should be taken to reduce the chances of biofilm formation on micro/nanorobots or related equipment. Biofilms can house microorganisms that may lead to colonization and potential harm to cells or embryos.

#### 2.7.4. Genetic and Epigenetic Considerations

Micro/nanorobots need to steer off-target effects that could cause mutations or changes in gene expression. Any unintentional genetic alterations could affect the health [9,17,82,85,86], development, or viability of embryos or future offspring. It is further important to understand and address any changes that could result from micro/nanorobotic interventions. These modifications can impact gene activity without altering the DNA sequence itself. It is crucial to study their effects on reproductive outcomes.

#### 2.7.5. Upholding Regulations and Ethics

Setting up safety standards, quality controls, and regulatory frameworks is vital for developing, testing, and using micro/nano technologies in IVF. Adhering to regulations ensures that patient safety remains a priority throughout the process. Also, patients undergoing IVF treatments involving micro/nanorobotics should be fully informed about the technology, its potential risks, benefits, and limitations, and alternative options. Obtaining consent allows patients to make choices about their reproductive health.

#### 2.7.6. Ensuring Continuous Monitoring

It is essential to keep track of patients who undergo IVF procedures with interventions over the long term. This monitoring helps evaluate outcomes, overall health status, and any potential effects on fertility or future offspring. Collecting data to assess the safety and effectiveness of/assisted IVF procedures plays a crucial role in analyzing and refining techniques and enhancing patient care and outcomes.

The key to a successful IVF procedure following all the safety measures is collaboration across interdisciplinary technical, ethical, and regulatory fields. Researchers, clinicians, bioengineers, ethicists, and regulatory bodies must work together to tackle the challenges associated with using/nanorobotic technologies in IVF. Providing education and training to healthcare professionals involved in these procedures is vital for maintaining competency following safety protocols and upholding standards. In summary, although the field of micro/nanorobotics holds promise in enhancing IVF procedures, it is crucial to assess the associated risks and implement safety measures. Factors such as biocompatibility, precision, infection prevention, genetic preservation, adherence to regulations, informed consent, and continuous monitoring play a role in ensuring the ethical application of micro/nanorobotic technologies in assisted reproduction methods like IVF. Collaboration, education, and a patient-centric approach are elements in addressing these complexities and maximizing the advantages of advancements in reproductive healthcare.

## 3. Future Directions

Looking toward the future, micro/nanorobotics stand poised to transform ARTs, promising to enhance the efficiency and success rates of these treatments. Continued advancements in micro/nanorobotics technology, characterized by miniaturization, integration of sensors and actuators, and automation, hold the key to unlocking more precise and efficient embryo manipulation and culture processes. The ongoing evolution of these robotics systems will enable researchers and clinicians to exert greater control over microenvironments within which embryos develop, optimizing conditions for embryo growth and maturation. This heightened level of precision in embryo manipulation has the potential to significantly improve the overall success rates of ART, offering renewed hope to individuals and couples grappling with fertility challenges.

The integration of AI and machine learning algorithms represents a notable shift in the process of selecting embryos and forecasting pregnancy outcomes based on embryo features (refer to Table 1). AI systems can identify complex patterns and biomarkers associated with embryo quality and developmental potential by analyzing data from microfluidic devices, imaging technologies, and genetic studies [44,86,87,88,89]. Prediction models and AI-driven decision-support systems let embryologists make more educated and personalized decisions when selecting embryos. The integration of micro/nanorobotics with AI technology has the potential to significantly enhance the effectiveness of ART, leading to improved pregnancy outcomes for individuals undergoing fertility treatment.

The integration of micro/nanorobotics with AI and machine learning can noticeably increase the effectiveness of ART, and pregnancy outcomes for people present process fertility remedies. Micro/nanorobotics could, in the future, perform specific responsibilities including biopsy, sperm injection, and embryo manipulation with high throughput accuracy and repeatability, lowering human mistakes and improving achievement prices. AI algorithms can optimize the control and coordination of those robot structures, enabling real-time monitoring and adaptive adjustments throughout processes. This interaction between micro/nanorobotics and AI advances the capabilities of ART. However, it also opens new avenues for studies and innovation within the discipline of reproductive medicine.

Furthermore, current studies in tissue engineering and organ-on-a-chip technologies are set to enhance the capacities of microfluidic systems utilized in ART [18,34,37,51,76]. Sophisticated microfluidic technologies have the potential to greatly enhance embryo culture settings in vitro by closely replicating the intricate in vivo microenvironments where embryos normally develop. Advanced microfluidic devices can enhance reproductive results in IVF treatments by creating a more biologically accurate environment for embryo growth. The combination of micro/nanorobotics with tissue engineering and organ-on-a-chip technologies shows great potential for enhancing ART. This integration could lead to improved success rates and overall experience for individuals and couples seeking fertility treatment.

### Summary of Our Current Research Pursuits

Our earlier research was primarily focused on automating the intricate 3D rotation process of individual oocytes through the innovative electrorotation method [90]. By meticulously generating a precise 3D electro-rotational field within a specialized microdevice, we achieved the advancement in rotating single bovine oocytes with unparalleled precision. Moreover, our study introduced a significant approach for evaluating cell health by analyzing rotation spectra, thus contributing valuable insights to the field [90].

Following our work on a single oocyte manipulation, our research trajectory shifted toward the manipulation and fabrication of magnetic bead swimmers that closely mimic the characteristics of sperm cells. This endeavor enabled us to achieve precise control over small particles and adeptly transport them to predefined target regions [91,92]. This phase of our research marked a significant advancement in the realm of microscale robotics and targeted delivery systems, with wide-ranging implications for various biomedical applications.

Presently, our research endeavors are directed towards evaluating the efficacy of microswimmers or micro/nanorobotic technology in targeting rotatable cells within 3D environments. Leveraging cutting-edge image processing and AI technologies, we aim to streamline and enhance the efficiency of high-throughput automation in this domain. Ultimately, our ongoing efforts strive to make substantial contributions to the fields of IVF and ART, promising transformative advancements in reproductive healthcare.

## 4. Conclusions

Micro/nanorobotics are ready to revolutionize IVF by improving the levels of precision, safety, and effectiveness in ART. Advanced technology can transform different areas of IVF such as focused sperm manipulation, less invasive oocyte retrieval, and precision embryo culture. This has the potential to improve IVF success rates and enhance patient outcomes. Current research and development in micro/nanorobotics are crucial for overcoming hurdles and preparing for the widespread use of micro/nanorobotic-assisted IVF in clinical settings, leading to advancements in reproductive healthcare. Progress in micro/nanorobotics can improve current IVF methods and introduce new approaches to boost the efficiency and success of ART. Researchers are advancing micro/nanorobotic systems for use in IVF, aiming to enhance outcomes for individuals and couples undergoing fertility treatments.

## Figures and Tables

**Figure 1 micromachines-15-00510-f001:**
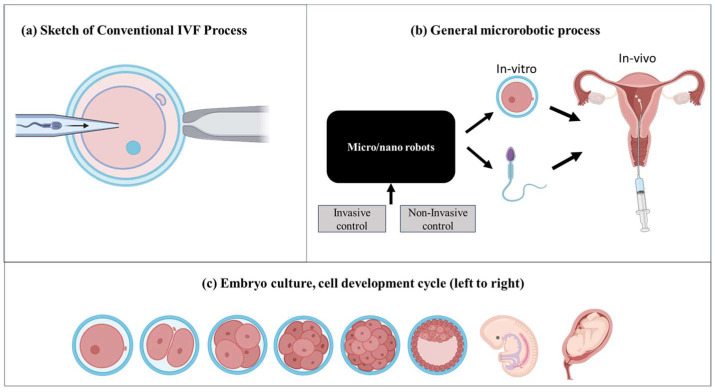
In vitro fertilization technique: (**a**) conventional IVF procedure where sperm is injected inside the egg cell or oocyte (**b**) micro/nanorobotic-based manipulation of the egg cell and sperm. Later, the fused cell with sperm is injected inside the uterine cavity; (**c**) and the embryo development cycle after it is injected inside the uterine cavity. Figure created with BioRender.com (accessed on 10 February 2024).

**Figure 2 micromachines-15-00510-f002:**
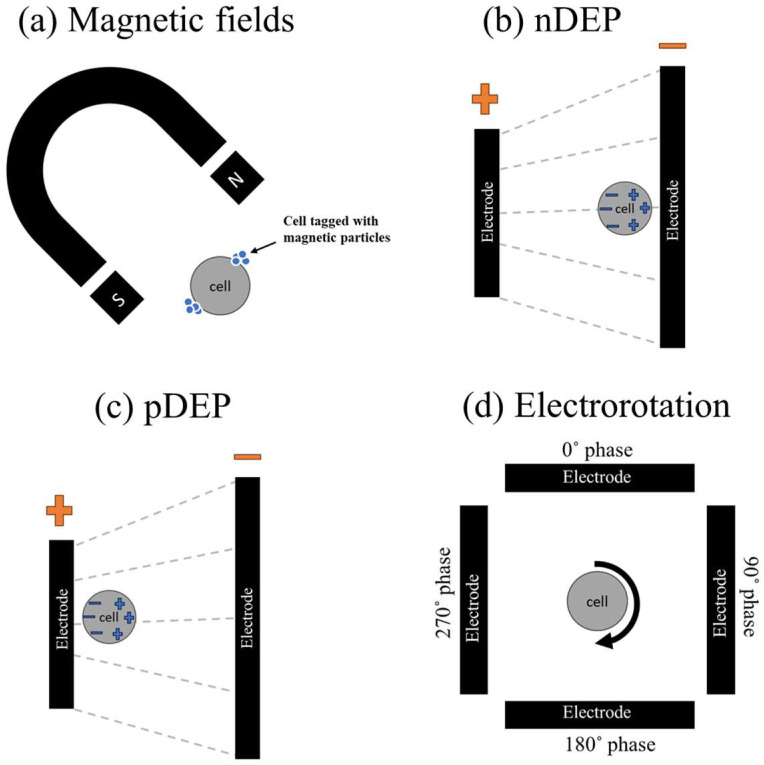
Gamete and embryo manipulation strategies. (**a**) Electromagnetic field manipulation of cells, (**b**) positive dielectrophoresis (pDEP), (**c**) negative dielectrophoresis (nDEP), and (**d**) electrorotation (ER).

**Figure 3 micromachines-15-00510-f003:**
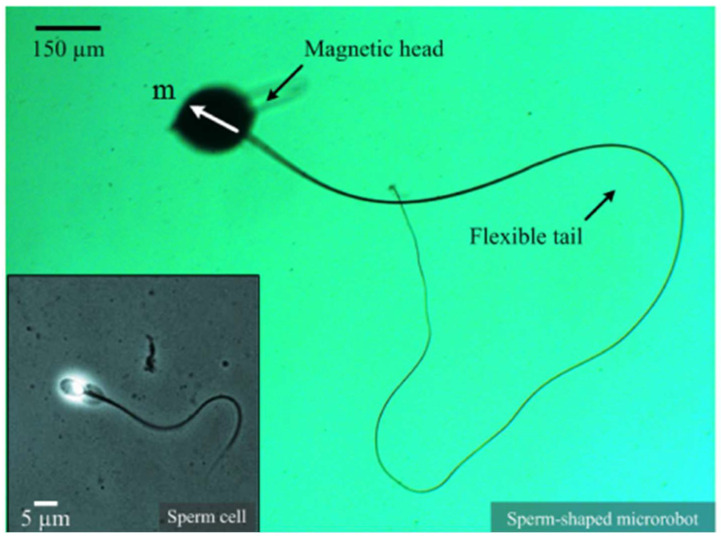
A microrobot with a form like that of a sperm is created using the process of electrospinning. This microrobot comprises a microbead and an ultra-fine fiber that closely match the physical structure of a sperm cell. The microbead is composed of iron oxide nanoparticles (45-00-252 Micromod Partikeltechnologie GmbH, Rostock, Germany) and possesses a magnetic dipole moment (m). On the other hand, the fiber generates propulsive force when subjected to oscillating magnetic fields. The fields are produced by arranging electromagnetic coils in an orthogonal configuration (as shown in the bottom-right inset) [43]. Figure 3 Reprinted with permission from Ref. [43]. Copyright 2016, IEEE.

**Figure 4 micromachines-15-00510-f004:**
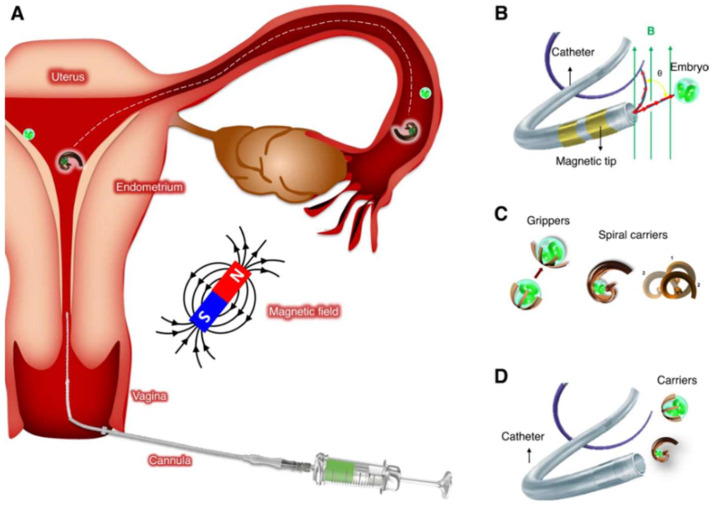
(**A**) A schematic representation of a micromotor resembling a spiral, which functions to capture, transport, and discharge an oocyte or embryo within the endometrium and fallopian tube. An overview of embryo transfer strategies: (**B**) tethered methodology incorporating a microcatheter; (**C**) untethered methodology utilizing microcarriers; and (**D**) a combined methodology in which untethered carriers are introduced via a catheter. © Reprinted/adapted with permission from Ref [7]. 9 February 2023, Springer Nature) with an open access article distributed under the Creative Commons Attribution License which permits unrestricted use, distribution, and reproduction in any medium.

**Figure 5 micromachines-15-00510-f005:**
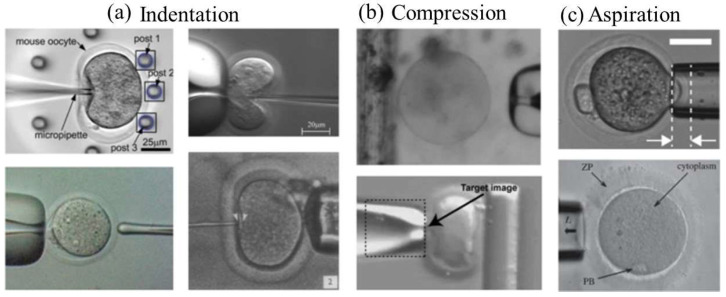
Micro-pipette techniques have been widely used to study the mechanical characteristics of oocytes and embryos, utilizing several methods for assessment. These approaches can be classified into three main types. (**a**) Two studies, Liu et al. (2012) [54] (top left) and Sun et al. (2003) [55], (top right) provide information on indentation-based techniques where an oocyte is either pushed against flexible supports or deformed by a force-sensing microneedle. Green (1987) [56] (bottom left) illustrates an oocyte being crushed by a quartz-fiber ‘poker’, while Murayama et al. (2004) [57] (bottom right) show an oocyte being probed by a material testing system (MTS) from the left side. The value ‘2’ on figure indicates the second figure [57] where in a Hamster egg zona was subjected to a compression force of 130 nN. (**b**) Compression-based methods: Abadie et al. (2014) [58] depict an oocyte about to be compressed between a micropipette and a floating platform, whereas Wacogne et al. (2008) [59] show an oocyte being compressed between a micropipette and a flexible post from a side angle. ‘Target image” refers to the algorithm used to monitor the movement of the pipette. (**c**) Aspiration-based methods: Yanez et al. (2016) [60] display a picture of an embryo being partially drawn into a micropipette, with the depth of aspiration marked between arrows. Khalilian et al. (2010b) [61] illustrate a portion of the zona pellucida (ZP) being drawn into a micropipette. Figure 3 Reprinted/adapted with permission from Ref. [53], Copyright 2016, Oxford University Press on behalf of the European Society of Human Reproduction and Embryology.

**Figure 6 micromachines-15-00510-f006:**
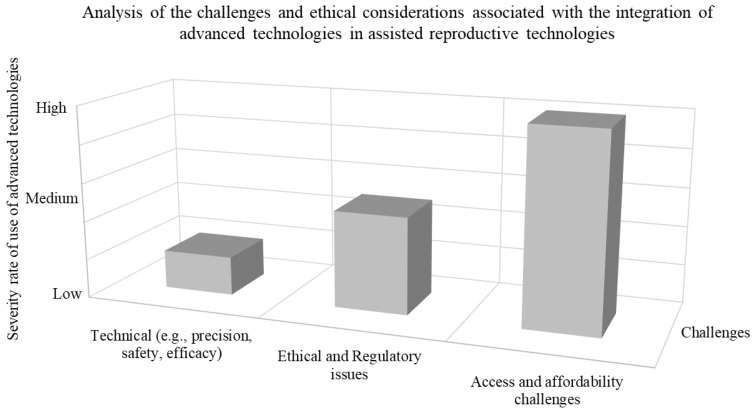
Analysis of the challenges and ethical considerations associated with the integration of advanced technologies into ART. The analysis data of the number of scientific publications indicate that there are still technical challenges to overcome. Some of the keywords used during a search in Web of Science and Scopus include: “advanced technologies during IVF and ART, Technical Issues in IVF, advanced technology in ARTs, Robotics in IVF and ART, Advanced technology ethical and regulatory issues in IVF, Affordability of use of advanced technologies in IVF”.

**Table 1 micromachines-15-00510-t001:** A few prominent works on AI and MI integration with micro/nanorobotics.

Title of Article	Journal Name	Publication Year
Autonomous 3D positional control of a magnetic microrobot using reinforcement learning [64].	*Nature Machine Intelligence*	2024
Application of micro/nanorobot in medicine [66].	*Frontiers in bioengineering and biotechnology*	2024
The prospect of artificial intelligence to personalize assisted reproductive technology [67].	*Digital Medicine*	2024
State of the Art in Actuation of Micro/Nanorobots for Biomedical Applications [68].	*Small Science*	2024
The Role of Artificial Intelligence and Machine Learning in Assisted Reproductive Technologies [69].	*Obstetrics and Gynecology Clinics of North America*	2023
Medical microrobots in reproductive medicine from the bench to the clinic [7].	*Nature Communications*	2023
Control and Autonomy of Microrobots: Recent Progress and Perspective [39].	*Advanced Intelligent Systems*	2022
Robotics, microfluidics, nanotechnology, and AI in the synthesis and evaluation of liposomes and polymeric drug delivery systems [65].	*Drug delivery and translational research*	2022
The Future Is Coming: Artificial Intelligence in the Treatment of Infertility Could Improve Assisted Reproduction Outcomes—The Value of Regulatory Frameworks [70].	*Diagnostics*	2021
Emerging Application of Nanorobotics and Artificial Intelligence To Cross the BBB: Advances in Design, Controlled Maneuvering, and Targeting of the Barriers [71].	*ACS Chemical Neuroscience*	2021
Recent progress in actuation technologies of micro/nanorobots [72].	*Beilstein journal of nanotechnology*	2021
Medical imaging of microrobots: toward in vivo applications [73].	*Medical Imaging of Microrobots: Toward In Vivo Applications*	2020
Sperm Cell Driven Microrobots—Emerging Opportunities and Challenges for Biologically Inspired Robotic Design [19]	*Micromachines*	2020
Machine learning for active matter [19].	*Nature Machine Intelligence*	2019
Artificial intelligence in reproductive medicine [63].	*Reproduction*	2019
Toward Better Treatment for Women’s Reproductive Health: New Devices, Nanoparticles and Even Robotic Sperm May Hold the Key to Preventing a Range of Health Conditions [74].	*IEEE pulse*	2018
Medical microbots need better imaging and control [75].	*Nature*	2017

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
