# Peer review of "Micro/Nanorobotics in In Vitro Fertilization: A Paradigm Shift in Assisted Reproductive Technologies"

_micromachines, 2024, doi:10.3390/mi15040510_

Round 1

Reviewer 1 Report

Comments and Suggestions for Authors

The paper provides a concise overview of the utilization of micro/nanorobotics in in-vitro fertilization (IVF), highlighting its potential to address various challenges associated with traditional IVF methods. The author has summarized the micro/nano robotic application in sperm manipulation, Oocyte retrieval and Embroyo culture.

There are one minor comment, the author called the micro pipette technique as microfluidic device seem not very correct, which is misleading with the microfluidic chip.

Comments on the Quality of English Language

No

Author Response

The author expresses gratitude to the reviewer for their insightful feedback and time dedicated to reviewing the article. I appreciate the feedback and have made the necessary correction to the caption on line 453 (current line 459) of Figure 5, replacing "Microfluidic devices" with "Micro pipette". Furthermore, the term "Microfluidic devices" has been revised to "Micro pipette" on line 502 (current line 528).

Reviewer 2 Report

Comments and Suggestions for Authors

This review paper presents the utilization of micro/nanorobotics in in-vitro fertilization (IVF) and its potential to revolutionize assisted reproductive technologies. It discusses the biomechanical characteristics of reproductive cells and their impact on various phases of the ART process, as well as the potential of microfluidic-based biomechanical analysis to enhance the selection of high-quality oocytes and embryos for ART procedures. The integration of micro/nano robotics with AI and ML algorithms.

Concerns in the paper include:

(1) It is necessary to further explore the challenges and limitations associated with the integration of micro/nanorobotics in IVF, as well as the ethical considerations and regulatory factors related to the use of advanced technologies in assisted reproductive technologies.

(2) The discussion AI and ML algorithms is not sufficient. There are many prominent reports about the integration of AI, ML and micro/nano robotics used in the topics discussed in the paper.

(3) The paper could benefit from a more comprehensive discussion of the potential risks and safety considerations associated with the use of micro/nanorobotics in IVF procedures.

(4) As a review paper, the abstract should be rewritten, and make it comply with the contents of the paper.

Overall, the paper provides insights into the potential impact of micro/nanorobotics on IVF processes and patient outcomes, but it would benefit from a more thorough analysis of the challenges and ethical considerations associated with the integration of advanced technologies in assisted reproductive technologies.

Round 2

Reviewer 2 Report

Comments and Suggestions for Authors

The draft has been improved, and everything looks fine.